# Skin Microbiome Overview: How Physical Activity Influences Bacteria

**DOI:** 10.3390/microorganisms13040868

**Published:** 2025-04-10

**Authors:** Cristina Mennitti, Mariella Calvanese, Alessandro Gentile, Aniello Vastola, Pietro Romano, Laura Ingenito, Luca Gentile, Iolanda Veneruso, Carmela Scarano, Ilaria La Monica, Ritamaria Di Lorenzo, Giulia Frisso, Valeria D’Argenio, Barbara Lombardo, Olga Scudiero, Raffaela Pero, Sonia Laneri

**Affiliations:** 1Department of Molecular Medicine and Medical Biotechnologies, Federico II University, Via Sergio Pansini 5, 80131 Napoli, Italy; cristinamennitti@libero.it (C.M.); mariellacalvanese99@gmail.com (M.C.); alexgenti98@libero.it (A.G.); vastola92@gmail.com (A.V.); pietromano97@gmail.com (P.R.); venerusoi@ceinge.unina.it (I.V.); scaranoc@ceinge.unina.it (C.S.); gfrisso@unina.it (G.F.); barbara.lombardo@unina.it (B.L.); olga.scudiero@unina.it (O.S.); 2Integrated Department of Laboratory and Transfusion Medicine, University of Naples Federico II, 80131 Naples, Italy; lauraingenito@yahoo.it (L.I.); luca.gentile@unina.it (L.G.); 3CEINGE-Biotecnologie Avanzate Franco Salvatore, Via G. Salvatore 486, 80145 Naples, Italy; lamonica@ceinge.unina.it (I.L.M.); dargenio@ceinge.unina.it (V.D.); 4Department of Pharmacy, University of Naples Federico II, Via Montesano, 80138 Naples, Italy; ritamaria.dilorenzo@unina.it (R.D.L.); slaneri@unina.it (S.L.); 5Department of Human Sciences and Quality of Life Promotion, San Raffaele Open University, Via di Val Cannuta 247, 00166 Roma, Italy; 6Task Force on Microbiome Studies, University of Naples Federico II, 80100 Naples, Italy

**Keywords:** antimicrobial peptides, physical activity, skin microbiome

## Abstract

The skin cannot be considered as just a barrier that protects against physical, chemical, and biological damage; it is a complex and dynamic ecosystem that varies across lifespans. Interest in the relationship between physical activity and skin microbiota has grown significantly in recent years. The skin microbiota has a crucial role in skin functions and physiology, and an imbalance, known as dysbiosis, is correlated with several diseases, such as inflammatory bowel disease (IBD), infectious disease, obesity, allergic disorders, and type 1 diabetes mellitus. Among the causes of dysbiosis, the practice of physical exercise, especially in contact sports, including wrestling, artistic gymnastics, and boating, certainly represents a predisposing factor for infectious disease. This review aims to provide an overview of the skin microbiota and its regulation, focusing on interactions between physical exercise and skin microbiota, the antimicrobial peptides (AMPs) as regulators of skin microbiota, and the impact of probiotics supplementation on physical performance.

## 1. Introduction

The skin has long been considered a barrier for protective action against physical, chemical, and biological damage. The spread of the research in metagenomics has revealed that almost all human body niches host microbes, including the skin. Thus, being colonized by bacteria, fungi, and viruses, skin cannot be considered to be just a wall; it is a complex ecosystem whose composition depends on mutual adaptation with the host and can be influenced by environmental and nutritional stimuli [1]. As for other body sites, the skin microbiota is crucial to ensuring skin physiological functions because dysbiotic status has been described in several diseases. Characterizing the skin microbiome may be useful in these conditions and may clarify the complex interplay between the human host and its microbes (Figure 1).

The main bacterial phyla that make up the skin microbiome fall into four different phyla: *Actinobacteria*, *Firmicutes*, *Bacteroidetes*, and *Proteobacteria*. Distributions of bacteria change depending on the skin site’s physiology, with specific bacteria being associated with moist, dry, and sebaceous environmental conditions. In general, bacterial diversity seems to be lowest in sebaceous sites, suggesting that there is selection for specific subsets of organisms that can tolerate conditions in these areas. *Propionibacterium* spp. is the dominant organism in these and other sebaceous areas, which confirms that *Propionibacterium* spp. is a lipophilic resident of the pilosebaceous unit [2]. *Staphylococcus* and *Corynebacterium* spp. are the most abundant organisms colonizing moist areas, suggesting that these organisms prefer areas of high humidity [2,3]. *Staphylococci* occupy an aerobic niche on the skin and probably use the urea in sweat as a nitrogen source. *Corynebacteria* are highly fastidious and slow-growing organisms in culture, and, as such, their role as skin microorganisms has been underappreciated until recently. The most diverse skin sites are the dry areas, with mixed representations from the phyla *Actinobacteria* [2,3,4]. These sites include the forearm, buttock, and various parts of the hand.

However, it must be underlined that the skin microbiota is extremely variable and can be influenced by several factors, such as ethnicity, gender, delivery mode, age, occupation, diet, hygiene habits, skin product usage, medication usage, climate, pollution, geographical location, and UV exposure [5].

Despite these variables, across our lifespan, skin colonization is driven by nutrient availability and environmental conditions. Indeed, based on body features, the skin takes on four different environments, i.e., sebaceous, moist, dry, and foot, that drive specific microbial colonization in such different skin niches [6]. In particular, in the sebaceous sites (face, chest, and back), the production of lipid-rich sebum induces the colonization of lipophilic taxa, such as *Cutibacterium* and *Malassezia*; moist sites (groin, elbow crease, and axilla) are characterized by the presence of sweat glands that promote *Staphylococcus* spp. and *Corynebacterium* spp. colonization; and dry sites (abdomen, palms, and forearm) feature a great variability in taxa composition but a low microbial abundance [7] (Table 1).

The skin microbiome composition varies across different body sites due to differences in regional skin anatomy. Some skin regions, such as the groin, axillary vault, and web, are partially occluded. These regions are higher in temperature and humidity, encouraging the growth of microorganisms that thrive in moist conditions (for example, Gram-negative bacilli, *Coryneforms*, and *Staphylococcus aureus*). The density of sebaceous glands is another factor that influences the skin microbiota, depending on the region. Areas with a high density of sebaceous glands, such as the face, chest, and back, encourage the growth of lipophilic microorganisms (for example, *Propionibacterium* spp. and *Malassezia* spp.) [8]. Compared with other skin sites, arm and leg skin is relatively desiccated and experiences large fluctuations in surface temperature. These areas are found to harbor quantitatively fewer organisms than moist areas of the skin surface [9].

**Table 1 microorganisms-13-00868-t001:** Microbial composition of each area of skin (data pooled from Grice et al. [10]).

Skin sites and Physiology	Alpha Diversity	Beta Diversity	Microbial Composition
**Dry** (hypothenar palm, volar forearm)	High	High interpersonal variation	*Actinobacteria* (*Propionibacterium* 13% and *Corynebacterium* 15%) *Firmicutes*, *Proteobacteria* (41%) and *Bacteroidetes* (14%)
**Moist** (Nare, antecubital fossa, inguinal crease, popliteal fossa)	Low	Low	Colonized predominantly by *Firmicutes* like *Staphylococcus* (21%), *Corynebacterium* spp. (28%), and *Proteobacteria* (26%)
**Sebaceous** (cheek, glabella, external auditory canal, occiput, back)	Lower	Lower	Colonized predominantly by *Propionibacterium* spp. (46%) and *Staphylococcus* (16%)

While it is true that microbial colonization depends on the physiological properties of the skin in different body sites, it must be noted that the skin’s physiology varies over time according to individual maturation and development. This means that the skin microbiota must be considered as a dynamic community that varies across lifespans.

From a functional perspective, as mentioned above, the mutual relationship between the skin microbiota and the human host is required for human health [11].

Indeed, the skin microbiota has been adapted to metabolize several host molecules, including proteins, carbohydrates, and sebum lipids, and produces bioactive compounds, such as antimicrobial peptides, phenol-soluble modulins, and antibiotics [5]. These metabolic activities are required for skin microbiota functions: immunity and inflammation regulation, protection against pathogens colonization, immune system shaping and adaptation during early life, maintenance of skin integrity and functions, UV protection, wound healing, and odor production.

For instance, the skin microbiota can produce short-chain fatty acids (SCFAs) through the metabolic processes of commensal bacteria like *C. acnes* and *S. epidermidis*. These SCFAs influence the production of cytokines and T cell activation, regulate the immune system, and have anti-inflammatory properties that maintain the integrity of the skin barrier [5]. Additionally, the pH and hydration level of the skin are influenced by these acid metabolites in conjunction with other microbial metabolites, like lactic acid and other organic acids [12]. Skin pH is required for several important skin functions, including barrier permeability, desquamation, and antimicrobial activities; thus, a microbial alteration may affect them.

A given person’s skin contains microorganisms that most likely come from a variety of places, such as inanimate objects, people, pets, cosmetics, air, and water [13,14,15,16]. Our current understanding of the relative contributions from these potential sources is only beginning. Human–surface and human-human contacts have long been acknowledged in medical literature as potent vectors of microbial dispersal [17,18]. Shaking hands, as well as hand contact with other body parts and room surfaces, has been found in culture-based studies to be potent vectors of health service infections, including *Klebsiella* spp. and methicillin-resistant *S. aureus* (MRSA) [19,20]. Since human contact with surfaces—specifically, other people’s skin surfaces—has been demonstrated to transmit specific microbial taxa, it is conceivable that human-to-human interactions result in the sharing of skin microbial communities.

The health of athletes and the impact of professional factors on it is currently receiving a lot of attention worldwide. However, practically all athletes experience skin issues at some point in their athletic careers [21], particularly concerning infectious skin diseases [22]. Purulent inflammatory skin diseases are at the top of the general hierarchy of infectious pathology in athletes [23]. Athletes’ ability to perform at a high level can be negatively impacted by the development of pathological conditions that arise from cases of infectious skin diseases caused by microbial transmission during training or competition [24]. Fungal infections, such as ringworm; viral infections, such as herpes gladiatorum in wrestlers, which is caused by *HSV-1*; and bacterial infections, such as impetigo, caused by *Staphylococci* or *Streptococci*, including methicillin-resistant *S. aureus* (MRSA), are among the most common infections seen in athletes [25,26].

## 2. Interactions Between Physical Exercise and Skin Microbiota

Athletes involved in contact sports, such as judo, rugby, and wrestling, belong to the population most vulnerable to skin infections caused by bacteria, viruses, or fungi [27,28,29].

There are several mechanisms by which infectious agents may be spread among athletes. Direct contact transmission involves person-to-person contact in which infectious agents are physically transferred from one person to a susceptible host. Indirect contact transmission occurs when a susceptible host meets contaminated objects or fomites, such as equipment, towels, or clothing. A type of indirect contact is droplet transmission, which occurs when droplets containing infectious agents are generated through coughing, sneezing, or talking and are deposited on the host’s conjunctivae, nasal mucosa, or mouth after being propelled in the air a short distance [30].

In contact sports, athletes frequently suffer from skin diseases [27,31]. For instance, wrestlers are at higher risk of developing dermatophytosis (*Tinea corporis*, *Tinea pedis*), impetigo, and *HSV* infection [32,33]. Shifts in the skin microbiome can accompany this high susceptibility rate because these changes in the bacterial community may cause a predisposition to infection. Intense skin-to-skin contact creates perfect conditions for the transmission of infectious agents [34].

These infections, which occur because of the dysbiosis of the skin microbiota, can be transmitted from one athlete to another directly through contact or indirectly through contaminated objects, such as towels, mats, and equipment.

Sports equipment and the environment in which physical activity takes place, not just direct contact between athletes, can influence and alter the health of the skin microbiota. This can increase athletes’ susceptibility to skin infections [35].

The predominant genera identified on surfaces, such as benches, elliptical handles, floors, free weights, and mats, include *Pseudomonas* and *Acinetobacter*, followed by *Staphylococcus*, *Corynebacterium*, and *Micrococcus* [36]. Each surface within a facility where physical activity occurs harbors a distinct microbial community, and the surfaces that come in contact frequently with athletes’ skin exhibit more dynamic and diverse microbial communities characterized by a non-random distribution. Notably, the microbial diversity presents in gyms and on sports equipment is significantly shaped by the microbiomes of the athletes rather than being influenced by the geographical location of the facility [37]. For example, gym mats, which are frequently in contact with athletes’ hands and feet, are a source/reservoir of opportunistic pathogenic microorganisms, such as *Staphylococcus*, *Corynebacterium*, and *Enhydrobacter* [38].

Symptoms resulting from the infections commonly include lesions, blisters, or sores, leading to abstention from training and competition. For this reason, it is important to protect skin health and, consequently, the skin microbiota, as this can impact athletic recovery and performance. Additionally, it has been observed that maintaining healthy skin enhances athletes’ confidence, demonstrating a positive impact on physical and mental health [39].

Bacterial infections affect athletes through direct contact. However, some authors highlight the roles of mats and equipment in spreading infections [40]. The treatment of deep bacterial infections may be challenging due to much higher skin colonization rates with methicillin-resistant *S. aureus* among athletes [41]. Common viral infections that afflict athletes include those caused by the *Herpes simplex virus* (*HSV*), *Human papillomavirus* (*HPV*), and *Molluscum contagiosum virus* (MCV). *HSV* can occur in any location, especially in contact sports. *HSV* infection can be easily contracted, resulting in massive outbreaks during athletic training camps [42,43]. Herpetic keratitis (involvement of the cornea) can lead to scarring, and repeated lesions can lead to permanent corneal opacity, requiring a corneal transplant to maintain good vision. Retinal necrosis leading to blindness may also occur. Retinal necrosis due to *HSV* infection is the most common cause of blindness of contagious origin in the USA [44,45,46].

Athletes are at high risk of fungal infections caused by dermatophytes due to increased exposure to pathogens (e.g., swimmers through contact with water, wrestlers through using mats) and repeated exposure to mechanical factors (e.g., micro-injuries of the skin of runners’ feet). This infection has been widely reported in wrestlers and is often referred to as *Tinea gladiatorum*, *Trichophytosis gladiatorum*, and *Tinea corporis gladiatorum* [32,47,48,49]. *T. capitis* is less common in athletes. This condition mostly affects people who practice sports that entail close contact among the athletes and those using protective equipment, e.g., jockeys and hockey players [50,51,52]. Tinea versicolor, a condition caused by a fungus (*Malassezia furfur*), is another relatively frequent superficial skin infection [53]. Its occurrence is associated with individual predisposition, excessive sweating, humidity, and closefitting clothing.

Exposure to different sports environments revealed that skin microbiome composition was enriched with methicillin-susceptible *S. aureus* (MSSA) and contained methicillin-resistant *S. aureus* (MRSA) (football locker room and weight room). The fingertip location of *S. aureus* recovery from football players was significant when compared with both other locations in football players, fingertips in wrestlers, and the control group. All *S. aureus* isolates recovered from athletes in our study were resistant to clindamycin, which is often the drug of choice for soft tissue infections. Trimethoprim/sulfamethoxazole is another drug of choice for soft tissue infections, but half of the isolates we recovered were resistant to it. If the bacteria are resistant to the administered antibiotic, the result may be septicemia; therefore, the clinician should be careful to select an antibiotic to which the infection is susceptible [54,55,56,57].

Pathogenic bacteria are the most responsible for skin infections and soft tissue infections (SSTIs) in the athletic population and include *S. aureus. S. aureus* can cause several types of infections, especially in athletes involved in sports like judo and wrestling [58,59]. These bacterial infections, from superficial skin infections to more serious invasive infections, usually affect the nostrils, upper airways, digestive tract, skin, and genital mucosa. Recently, *S. aureus* infections have demonstrated resistance to methicillin or similar penicillin antibiotics, which make treatment and a cure difficult. These strains are referred to as MRSA. In addition to bacteria, viruses are also responsible for skin infections. Viral infections can occur in athletes, and in particular, verruca (warts), molluscum contagiosum, and herpes simplex are the most widespread. Swimmers could be particularly susceptible to the plantar verruca, and in this case, the only ways to prevent the spread of the wart are to treat it with liquid nitrogen and topical keratolytic/salicylic acid preparations and for the athletes to use sandals when using shared showers [60]. *Molluscum contagiosum* is characterized by discrete papules and can be found particularly in wrestlers. *M. contagiosum* is highly contagious through direct skin contact, so to avoid its spread among athletes, it must be treated promptly with destructive methods like liquid nitrogen [60]. *HSV* types I and 2 are common infectious agents in humans. The term *Herpes gladiatorum* (*HG*) became widely spread in 1989 when 60 wrestlers contracted the virus. It has been reported that 94% to 97% of *HSV* infections in wrestlers are caused by type 1 [61]. The timely identification and treatment of affected individuals and their exclusion from the activity to avoid contact with other wrestlers can avoid its spread.

Finally, fungal organisms can also significantly affect athletes. For example, tinea pedis can affect many athletes because its growth is favored by warm and humid environments. For this reason, runners, skaters, and long-distance walkers may be particularly at risk. Affected athletes should be treated with antifungal agents and wear shoes in shared spaces to reduce transmission [60].

Several studies have evaluated the correlation between physical exercise and the skin microbiome. A study conducted on 15 wrestlers, comparing the composition of the skin microbiome before and after training, demonstrated that the main cause of its alterations was not due to the sharing of training mats but to the transmission of microorganisms following physical contact between athletes [62]. A study by Meadow on flat-track roller derby players showed that their own skin microbial community composition characterized each team membership. This aspect could be because each team was from a different geographic area associated with a different climate, urban setting, outdoor microbiota, and also a very different environmental microbiota [31]. The most interesting aspect was that, following the competition, the players’ microbial communities became more like each other.

Physical exercise has been demonstrated to modulate immune function, thereby strengthening the body’s defenses against infections interacting with skin microbiota. By modulating the immune system and decreasing levels of pro-inflammatory cytokines, exercise has been observed to alleviate symptoms such as skin thickness and redness, which are common in inflammatory skin diseases.

While physical activity confers numerous health benefits, it also presents certain skin types, particularly in terms of increased susceptibility to infections [63].

The molecular mechanisms by which physical activity influences skin microbiota include improved blood flow to the skin, promotion of skin hydration, positive effects on the hormonal system, increased collagen production in the dermal layer, and positive influences on the mechanical properties of the skin [64]. Physical activity stimulates the release of interleukin-15, which prevents mitochondrial dysfunction and promotes mitochondrial biosynthesis. This, in turn, may correlate with better skin hydration in physically active individuals [65,66,67,68,69]. Proper skin hydration improves its natural barrier function, facilitating protection against internal and external irritants, thereby preventing the development of common skin conditions, such as atopic dermatitis, contact dermatitis, psoriasis, and acne [65,70,71].

The Centers for Disease Control and Prevention [72], the Infectious Diseases Society of America [73], and some athletic associations, such as the National Athletics Trainers Association [74] and National Collegiate Athletic Association [75], have all developed guidelines to avoid bacterial and fungal infections in locker rooms, private and school gyms, and fitness centers. The presence of open wounds, abrasions, or lacerations should be a reason for the athlete’s exclusion from common pools, and shared spaces should be sanitized after each use. Some studies also revealed that, in individuals with *S. aureus* SSTIs, the addition of chlorhexidine or immersion to normal routine hygiene measures in a bathtub containing a diluted solution of household bleach bubble bath can represent a valid tool for fighting microbial colonization and prevention. Adequate cleanliness and hygiene are a crucial point in this process; in fact, long pauses in competitive and training activities caused by infections can lead to a decline in athletic performance. For this, athletic trainers and sports doctors have an important role in educating athletes about preventing these types of infections [76].

## 3. Regulators of Skin Microbiota: Antimicrobial Peptides (AMPs)

The skin microbiome is an ecosystem consisting of multiple microbial species essential for the maintenance of skin physiology and immunity, with a crucial role in regulating its homeostasis. Dysbiosis of the microbiota can cause skin inflammation and the subsequent development of skin diseases. Therefore, there are specific mechanisms that control and shape the microbiota, enabling a proper balance of its composition [77]. An important regulatory role in the composition of the microbiome is played by host antimicrobial peptides (AMPs). Small peptides known as AMPs are widely expressed on the skin. They originate in the deeper layers of the epidermis and are then transported to the stratum corneum, where they function as multifunctional effector molecules that connect innate and adaptive immune responses, as well as serve as the body’s first line of defense against possible pathogens [78].

AMPs have a crucial role in regulating the skin microbiota. HBD-2 exhibits high activity against Gram-negative bacteria, such as *P. aeruginosa* and *E. coli*, as well as against the yeast *C. albicans* [79,80]. Another beta-defensin, hBD-3, has a very broad range of activity, and it acts in low doses and remains effective, even in high salt concentrations [81]. Also, the inducible AMP called CAMP (cathelicidin antimicrobial peptide) encodes for an 18 kDa precursor, which is proteolytically processed to a 37 amino acid-containing peptide, termed LL-37. LL-37 is the only human cathelicidin, and it is also expressed in the skin [82]. LL-37 shows broad antimicrobial activity against Gram-positive and Gram-negative bacteria and fungi, including yeasts [83,84], in contrast to hBD-2, hBD-3, and LL-37, which are only weakly expressed in healthy skin. Gläser et al. identified the S100 protein psoriasin (S100 A7) as an abundant AMP of healthy skin [85]. Psoriasin exhibits antimicrobial activity, especially against *E. coli*, and its reduced form has been reported to be active against various fungi [86]. Harder et al. isolated a new AMP from the stratum corneum of healthy skin with high antimicrobial activity against a wide range of microorganisms [87]. Because of its structural similarity, this AMP was assigned to the ribonuclease A superfamily and is referred to as ribonuclease 7 (RNase7). RNase 7 is a cationic, lysine-rich 14.5 kDa protein with a broad spectrum of antimicrobial activities and very potent ribonuclease activity. RNase 7 is abundant in human skin, and pro-inflammatory cytokines and bacteria can further induce its expression [87,88,89,90,91]. The inactivation of psoriasin on the skin surface enhances the growth of applied *E. coli* [85]. The antibody-based inactivation of RNase7 reveals its crucial role in controlling the cutaneous growths of *S. aureus*, *Corynebacterium amycolatum*, *E. faecium*, and *P. aeruginosa*. The antibody-based neutralization of hBD-3 reveals the important role of hBD-3 in restricting *S. aureus* growth [90,92,93,94].

Also, AMPs increase the production of chemokines and cytokines, draw immune cells to the infection site, alter the responses of Toll-like receptors, and bind and deactivate bacterial endotoxins, all of which promote wound healing and angiogenesis [95,96]. In addition to directly eliminating pathogens, AMPs regulate immune responses and interfere in cell differentiation, re-epithelialization, and their cooperative interactions with the skin microbiota.

The key regulatory mechanisms through which AMPs modulate skin immunity and prevent infections include: (1) protection from microbial infection (broad spectrum of antimicrobial activity against bacteria, yeast, fungi, protozoa, and viruses); (2) improvement of skin barrier homeostasis (regulation of the normal skin microflora composition); (3) modulation of inflammation responses (controlling the production of various cytokines/chemokines), and (4) promotion of wound healing. AMPs also initiate a potent host response to skin infection, resulting in cytokine/chemokine production, inflammation, and a cellular response [97,98]. After exposure to microbe-derived molecules, monocytes and lymphocytes stimulate the epidermal expression of hBDs [99]. The hBD-2, hBD-3, and hBD- 4, but not hBD-1, stimulate keratinocytes to produce proinflammatory cytokines and chemokines through the G protein-coupled receptor (GPCR) and phospholipase C (PLC) signaling pathways [100]. Furthermore, hBDs induce keratinocyte migration and proliferation, which involves EGFR, a signal transducer and activator of transcription (STAT)1 and STAT3 activation. Cathelicidin, LL-37, also synergizes with endogenous inflammatory mediators to enhance the induction of specific inflammatory effects through a complex mechanism involving multiple pathways, such as GPCR, EGFR, and TLR [101,102]. As a result, cathelicidin peptides increase cell migration and the secretion of cytokines (IL-6, IL-8, IL-10, IL-18, and IP-10) and chemokines (MCP-1, MIP3α, and RANTES) from activated cells. Also, dermcidin-1L (DCD-1L) stimulates keratinocytes to generate cytokines (TNF- α, IL-8, and IP-10) and chemokines (MIP3α) through both G protein and p38/MAPK pathways [103].

All of these factors are critical for preserving an ideal and functional skin barrier [104,105]. The antimicrobial capacity of AMPs, together with their epithelial expression, makes their roles as regulators and shapers of the microbiota likely, and the main classes are reported in Table 2. The most widely characterized families of AMPs in the skin are defensins and cathelicidins. Defensins are small cationic peptides; they have β-sheet structures with cysteine-rich residues that form characteristic disulfide bridges [106]. The main defensins are α- and β-defensins, according to the position of the disulfide bridges.

Human α-defensins (HNPs) are mainly produced in neutrophils. In particular, there is a high expression of HNP1-3 and more moderate activity of HNP4. Among them, HNP2 has important bactericidal activity against *S. aureus*, compared to HNP1, 3, and 4 [107].

Human β-defensins (hBDs) are released by epithelial cells, such as keratinocytes, activated monocytes/macrophages, and dendritic cells [107]. Among β-defensins, hBD-2 has bacteriostatic activity against Gram-negative bacteria, such as *P. aeruginosa* and *E. coli*, as well as against the yeast *C. albicans*, while hBD-3 is very potent against *C. albicans*, *E. coli*, *S. pyogenes*, *P. aeruginosa*, and *E. faecium*. In turn, hBD-2 and hBD-3 are induced by proinflammatory cytokines, such as interleukin-1β (IL-1β) and tumor necrosis factor (TNF-α)/interferon-γ (IFN-γ), but also by microbial, injury, and UV-B stimuli [101]. Cathelicidins, also called LL-37, are small, cationic, amphipathic peptides consisting of 12 to 80 amino acids; LL-37 is the only human cathelicidin also expressed on the skin and shows broad antimicrobial activity against Gram-positive and Gram-negative bacteria [78]. The skin defense barrier also includes other less common antimicrobial peptides found in skin cells, such as psoriasin, RNase7, and dermcidin.

When applied at lower concentrations, psoriasin, found in the keratinocytes of psoriasis patients, is an extremely effective AMP against *E. coli* and is active against *S. aureus* at higher concentrations [108]. It is upregulated in psoriasis and chronic wounds, and pro-inflammatory cytokines can induce its expression in keratinocytes, and it is a potent modulator of neutrophil activation [109]. RNase7, produced in keratinocytes, is involved in protecting the skin from infection caused by *S. aureus* and has antimicrobial activities against *E. coli*, *P. aeruginosa*, *E. faecium*, and *P. acnes* [52]. Finally, dermcidin is secreted by sweat glands and has antimicrobial activities against *S. aureus*, *E. coli*, and *C. albicans* [109].

Despite such evidence, the impact of skin-derived AMPs on the microbiota is yet to be defined, but there is growing evidence that AMPs can modulate and balance the microbiota. This hypothesis is based on the direct antimicrobial activity of AMPs against microbiota members and microbiota-regulated AMP expressions in the skin. Members of the skin microbiota and their secreted products detected by host keratinocytes and immune cells induce the upregulation of genes involved in the immune system and inflammatory response, including several AMPs, indicating that the microbiota contributes to providing the skin with constant AMP-mediated antimicrobial action. One study showed that the stimulation of keratinocytes with *S. epidermidis* or its secreted factors induces the expression of many AMPs, such as ß-defensins hBD-2 and hBD-3 or RNase7. S. epidermidis stimulation of nasal keratinocytes also increases the expressions of hBD-3, RNase 7, and LL-37. Interestingly, the inductions of hBD-3 and RNase 7 by *S. epidermidis* highlight a possible mechanism of how skin commensals, such as *S. epidermidis*, amplify the innate immune response in the presence of infection [110,111,112,113,114]. The Gram-negative mucosal bacterium *R. mucosa* is a member of the healthy skin microbiota, and its absence in atopic dermatitis (AD), a skin infection caused by *S. aureus*, can trigger the disease. Skin treatment of AD patients treated with R. mucosa reduces disease severity and the *S. aureus* burden. Interestingly, *R. mucosa* induces the expressions of hBD-2 and LL-37 in keratinocytes, suggesting that AMP induction could be used to treat AD patients [115,116]. The above studies document that members of the skin microbiota also induce the expressions of various AMPs. This points to the presence of a feedback loop that regulates the precise AMPs–microbiota relationship and homeostasis: the microbiota induces AMP expression, which leads to the better control of AMP-mediated growth of microbiota members and results in a decrease in microbiota abundance, followed by reduced AMP induction [109]. The skin microbiota could represent an interesting opportunity to develop new therapies to improve skin health and treat skin infections. It has been observed that skin diseases are associated with a reduced diversity of the microbiota with specific microorganisms; using a cocktail of different strains as therapy would better suit the personal pathological situation of the recipients, offering therapeutic advantages [109].

The applied microbiota can be: (1) alive (probiotics): a probiotic is a living microorganism that, when added in sufficient amounts, exerts a beneficial effect on the host [117]. (2) Tyndallized or thermos-killed bacteria (postbiotics): bacterial cell structures, enzymes and excreted bacterial factors are added, but the bacteria do not replicate anymore. (3) Cell lysates or physically killed bacteria (postbiotics): the bacteria are destroyed, and the cell contents and cell walls are in the solution. The bacteria do not replicate anymore, but the enzymes can still be active. (4) Purified enzymes: single or groups of bacterial enzymes are purified and added. (5) Fermentation products or supernatants: the bacteria are not added, but the supernatants containing their antioxidants, amino acids, lipids, and/or vitamins are added. Methods 1–5 have multiple advantages over a skin microbiome transplant, with the main advantage being that the process is easily scalable and thus industrially applicable. For method 1 (application of live probiotics), highly concentrated bacteria can be applied; thus, a higher efficacy can be obtained, compared to a complete skin microbiome transplant. Pro- or postbiotics can be applied in a skin emollient, cream, or suitable medium for skin. There are also a series of drawbacks associated with the use of pro- and postbiotics. Bacteria are cultured in sugar-rich media; it can therefore be more difficult for the bacteria to adjust to a sebum-rich environment. Skin engraftment is not easy; the applied bacteria compete with the skin resident microbiome of the deeper skin layers. The application of high amounts of bacteria could lead to a skin immune reaction, with irritation and side effects as a result. A third method of changing the skin microbiome is through prebiotic stimulation. In this process, prebiotics are supplemented to the skin to stimulate the growth of specific health-promoting microbes. A prebiotic is an ingredient with a bioselective activity that exerts a beneficial effect on the host and attempts to improve the host’s health [118]. There are several advantages to this method. There is no need to work with living bacteria; thus, there is a reduced chance of a skin immune reaction. The method has an indirect mechanism of action. Prebiotics are typically well-defined compounds for which side effects are well-studied. The INCI name and safety sheets are normally available. There are also disadvantages. The indirect method has less direct results. Prebiotics could also stimulate non-targeted, low-abundance bacteria. The effect of prebiotics can be unpredictable, given the variability in the skin microbiome, physiology, and immune response in different individuals. All methods have their advantages and disadvantages. Scientific research is currently being conducted using several of these methods to treat common skin disorders.

Paetzold et al., through their studies, have shown the possibility of developing specific probiotic solutions based on the healthy skin microbiota that can convert a diseased microbiota status to a healthier one in the recipient [119]. In addition to the application of skin-derived commensals, different studies have shown how the use of probiotics containing non-skin-derived probiotics, such as lactic acid bacteria, improves skin diseases, such as AD or acne vulgaris [120,121].

## 4. Conclusions

This review highlights the intricate relationship between physical activity and skin microbiota, outlining the advantages and potential obstacles related to it. The skin hosts an intricate community of microorganisms (the skin microbiome), which is preeminent for maintaining skin health and protecting against pathogens. The skin microbiome’s composition is influenced by host physiology, lifestyle, and environmental factors, such as personal hygiene, physical activity, climate, and geography. Microbes can be moved to the skin through contact with various surfaces, other people, pets, cosmetics, or the environment. Physical activity is effective in maintaining skin mechanics, promoting immune defense, and fostering skin microbiota vitality. Athletes face unique challenges that can change the microbial composition of their skin. These challenges include exposure to various terrains, harsh weather conditions, crowded living spaces, limited access to sanitation, field exercises, and their typical work environments. External factors can disrupt the skin’s microbial balance, leading to skin health issues and increased vulnerability to infections. Athletes frequently encounter skin and soft tissue infections (SSTIs) due to proximity and frequent physical contact.

The impact of skin health not only affects individuals but also places a financial strain on the healthcare system. This highlights the importance of understanding and addressing the factors that influence the skin microbiome, which plays a very important role in preventing infections and maintaining overall health during physical activities. By studying the modulation of the skin microbiome, researchers can develop improved hygiene protocols and skincare strategies that enhance the health and well-being of athletes and others in demanding environments. Additionally, these insights can contribute to better infection control and hygiene practices beyond the realm of athletics.

## Figures and Tables

**Figure 1 microorganisms-13-00868-f001:**
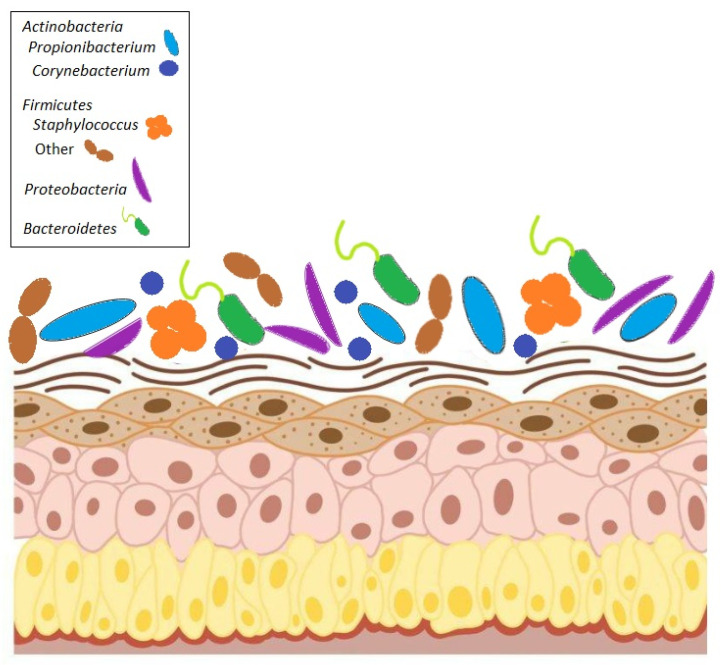
Composition of human skin microbiome. The microbiome consists of four major phyla: *Actinobacteria*, *Firmicutes*, *Bacteroidetes*, and *Proteobacteria*.

**Table 2 microorganisms-13-00868-t002:** The main AMPs of the skin and their major targets.

Antimicrobial Peptides of the Skin	Bacteriostatic Activity Against
*HNP2*	Gram-positive (*S. aureus*)
*hBD2*	Gram-negative (*P. aeruginosa* and *E. coli*), Gram-positive (*S. epidermidis*), and yeast (*C. albicans*)
*hBD3*	Gram-negative (*E. coli* and *P. aeruginosa*), Gram-positive (*S. pyogenes*, *E. faecium*, *S. aureus*, and *S. epidermidis*), and yeast (*C. albicans*)
*LL-37*	Gram-positive (*S. epidermidis*) and Gram-negative
*Psoriasin*	Gram-negative (*E. coli*) and Gram positive (*S. aureus*)
*RNAase7*	Gram-negative (*P. aeruginosa* and *E. coli*) and Gram-positive (*S. aureus*, *C. amycolatum*, *E. faecium*, and *P. acne*)
*Dermcidin*	Gram-negative (*E. coli*), Gram-positive (*S. aureus*), and yeast (*C. albicans*)

## Data Availability

No new data were created or analyzed in this study.

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
