# Peer review of "Skin Microbiome Overview: How Physical Activity Influences Bacteria"

_microorganisms, 2025, doi:10.3390/microorganisms13040868_

Round 1
Reviewer 1 Report
Comments and Suggestions for Authors
Minor comments:
How does the skin microbiome composition vary across different body sites, and what factors influence these variations? (Lines 36–64)
What are the main bacterial phyla that make up the skin microbiome, and how do their distributions change under different environmental conditions? (Lines 48–49)
How does physical activity, particularly contact sports, contribute to skin microbiome dysbiosis and the spread of infections? (Lines 201–218)
What are the roles of antimicrobial peptides (AMPs) in regulating the skin microbiota, and how do they interact with different microbial species? (Lines 113–132)
How do common bacterial, viral, and fungal infections affect athletes, and what are the primary risk factors for these infections? (Lines 227–252)
What are the key regulatory mechanisms through which antimicrobial peptides (AMPs) modulate skin immunity and prevent infections? (Lines 120–129)
How does direct and indirect microbial transmission occur among athletes, and what preventive measures can be implemented? (Lines 203–214)
What impact does exposure to different sports environments (e.g., gym mats, locker rooms) have on the composition and resilience of the skin microbiome? (Lines 208–219)
What are the benefits and limitations of using probiotic and prebiotic interventions to restore and maintain a healthy skin microbiome? (Lines 189–198)
How does physical exercise modulate the immune response through changes in the skin microbiota, and what implications does this have for infection susceptibility? (Lines 253–263)
Author Response
Thank you very much for taking the time to review this manuscript. Please find the detailed responses below and the corresponding revisions/corrections highlighted/in track changes in the re-submitted files.
Comment 1: How does the skin microbiome composition vary across different body sites, and what factors influence these variations? (Lines 36–64)
Response 1: Thank you for point this out. We agree with this comment.
The skin microbiome composition varies across different body sites due to differences in regional skin anatomy. Some skin regions, such as the groin, axillary vault, and web, are partially occluded. These regions are higher in temperature and humidity, encouraging the growth of microorganisms that thrive in moist conditions (for example, Gram-negative bacilli, Coryneforms, and Staphylococcus aureus). The density of sebaceous glands is another factor that influences the skin microbiota, depending on the region. Areas with a high density of sebaceous glands, such as the face, chest, and back, encourage the growth of lipophilic microorganisms (for example, Propionibacterium spp. and Malassezia spp.) [8]. Compared with other skin sites, arm and leg skin is relatively desiccated and experiences large fluctuations in surface temperature. These areas were found to harbor quantitatively fewer organisms than moist areas of the skin surface [9].
Comment 2: What are the main bacterial phyla that make up the skin microbiome, and how do their distributions change under different environmental conditions? (Lines 48–49)
Response 2: Thank you for point this out. We agree with this comment.
The main bacterial phyla that make up the skin microbiome fall into four different phyla: Actinobacteria, Firmicutes, Bacteroidetes, and Proteobacteria. Distributions of bacteria change is dependent on the physiology of the skin site, with specific bacteria being associated with moist, dry, and sebaceous environmental conditions. In general, bacterial diversity seems to be lowest in sebaceous sites, suggesting that there is selection for specific subsets of organisms that can tolerate conditions in these areas. Propionibacterium spp. are the dominant organisms in these and other sebaceous areas, which confirms that Propionibacterium spp. is a lipophilic resident of the pilosebaceous unit [2]. Staphylococcus and Corynebacterium spp. are the most abundant organisms colonizing moist areas, suggesting that these organisms prefer areas of high humidity [2,3]. Staphylococci occupy an aerobic niche on the skin and probably use the urea present in sweat as a nitrogen source. Corynebacteria are extremely fastidious and slow-growing organisms in culture, and, as such, their role as skin microorganisms has been underappreciated until recently. The most diverse skin sites are the dry areas, with mixed representation from the phyla Actinobacteria [2-4]. These sites include the forearm, buttock, and various parts of the hand.
Comment 3:
How does physical activity, particularly contact sports, contribute to skin microbiome dysbiosis and the spread of infections? (Lines 201–218)
Response 3: Thank you for point this out. We agree with this comment.
There are several mechanisms by which infectious agents may be spread among athletes. Direct contact transmission involves person-to-person contact in which infectious agents are physically transferred from one person to a susceptible host. Indirect-contact transmission occurs when a susceptible host meets contaminated objects or fomites, such as equipment, towels, or clothing. A type of indirect contact is droplet transmission, which occurs when droplets containing infectious agents are generated through coughing, sneezing, or talking and are deposited on the host’s conjunctivae, nasal mucosa, or mouth after being propelled in the air a short distance. In contact sports, athletes frequently suffer from skin diseases [27,30]. For instance, wrestlers are at higher risk of developing dermatophytosis (Tinea corporis, Tinea pedis), impetigo, and herpes simplex virus infection [31,32]. Shifts in the skin microbiome can accompany this high susceptibility rate because these changes in the bacterial community may cause a predisposition to infection. Intense skin-to-skin contact creates perfect conditions for the transmission of infectious agents.
Comment 4:
What are the roles of antimicrobial peptides (AMPs) in regulating the skin microbiota, and how do they interact with different microbial species? (Lines 113–132)
Response 4: Thank you for point this out. We agree with this comment.
AMPs have a crucial role in regulating the skin microbiota. HBD-2 exhibited high activity against Gram-negative bacteria such as P. aeruginosa and E. coli, as well as against the yeast C. albicans [72,73] Another beta-defensin, hBD-3, has a very broad range of activity, and it acts in low doses and remains effective even in high salt concentrations [74]. Also, the inducible AMP called CAMP (cathelicidin antimicrobial peptide) encodes for an 18 kDa precursor, which is proteolytically processed to a 37 amino acid-containing peptide, termed LL-37. LL-37 is the only human cathelicidin, and it is also expressed in the skin [75] LL-37 shows broad antimicrobial activity against Gram-positive and Gram-negative bacteria and fungi, including yeasts [76,77]. In contrast to hBD-2, hBD-3, and LL-37, which are only weakly expressed in healthy skin. Gläser et al. identified the S100 protein psoriasin (S100 A7) as an abundant AMP of healthy skin [78]. Psoriasin exhibits antimicrobial activity, especially against E. coli, and its reduced form has been reported to be active against various fungi [79]. Harder et al. isolated a new AMP from the stratum corneum of healthy skin with high antimicrobial activity against a wide range of microorganisms. Because of its structural similarity, this AMP was assigned to the ribonuclease A superfamily and referred to as ribonuclease 7 (RNase7) [80]. RNase 7 is a cationic, lysine-rich 14.5 kDa protein with a broad spectrum of antimicrobial activity and very potent ribonuclease activity. RNase 7 is abundant in human skin, and pro-inflammatory cytokines and bacteria can further induce its expression [80-84]. The inactivation of psoriasin on the skin surface enhanced the growth of applied E. coli [78]. Antibody-based inactivation of RNase7 revealed its crucial role in controlling the cutaneous growth of S. aureus, Corynebacterium amycolatum, E. faecium and P. aeruginosa. Antibody-based neutralization of hBD-3 revealed an important role of hBD-3 in restricting S. aureus growth [83,85-87].
Comment 5:
How do common bacterial, viral, and fungal infections affect athletes, and what are the primary risk factors for these infections? (Lines 227–252)
Response 5: Thank you for point this out. We agree with this comment.
Bacterial infections affect athletes through direct contact. However, some authors highlight the role of mats and equipment in spreading infections [35]. The treatment of deep bacterial infections may be challenging due to much higher skin colonization rates with methicillin-resistant S. aureus among athletes [36]. Common viral infections that afflict athletes include those caused by Herpes simplex virus (HSV), Human papillomavirus (HPV), and Molluscum contagiosum virus (MCV). Herpes simplex virus can occur in any location, especially in contact sports. The HSV infection can be easily contracted, resulting in massive outbreaks during athletic training camps [37,38]. Herpetic keratitis (involvement of the cornea) can lead to scarring, and repeated lesions can lead to permanent corneal opacity, requiring a corneal transplant to maintain good vision. Retinal necrosis leading to blindness may also occur. Retinal necrosis due to HSV infection is the most common cause of blindness of contagious origin in the USA [39-41].
Athletes are at high risk of fungal infections caused by dermatophytes due to increased exposure to pathogens (e.g., swimmers through contact with water, wrestlers through using mats) and repeated exposure to mechanical factors (e.g., micro-injuries of the skin of runners’ feet). This infection has been widely reported in wrestlers and is often referred to as Tinea gladiatorum, Trichophytosis gladiatorum and Tinea corporis gladiatorum [31, 42-44]. Tinea capitis is less common in athletes. This condition mostly affects people who practice sports that entail close contact among the athletes and those using protective equipment, e.g., jockeys and hockey players [45-47]. Tinea versicolor, a condition caused by a fungus (Malassezia furfur), is another relatively frequent superficial skin infection [48]. Its occurrence is associated with individual predisposition, excessive sweating, humidity, and closefitting clothing.
Comment 6:
What are the key regulatory mechanisms through which antimicrobial peptides (AMPs) modulate skin immunity and prevent infections? (Lines 120–129)
Response 6: Thank you for point this out. We agree with this comment.
The key regulatory mechanisms through which AMPs modulate skin immunity and prevent infections include: 1) protection from microbial infection (broad spectrum of antimicrobial activity against bacteria, yeast, fungi, protozoa, and viruses); 2) improvement of skin barrier homeostasis (regulation of the normal skin microflora composition), (3) modulation of inflammation responses (controlling the production of various cytokines/chemokines), and (4) promotion of wound healing. AMPs also initiate a potent host response to skin infection, resulting in cytokine/chemokine production, inflammation, and a cellular response [90,91]. After exposure to microbe-derived molecules, monocytes and lymphocytes stimulated the epidermal expression of hBDs [92]. The hBD-2, hBD-3, and hBD- 4 but not hBD-1 stimulate keratinocytes to produce proinflammatory cytokines and chemokines through the G protein-coupled receptor (GPCR) and phospholipase C (PLC) signaling pathways [93]. Furthermore, hBDs induce keratinocyte migration and proliferation, which involves EGFR, a signal transducer and activator of transcription (STAT)1 and STAT3 activation. Cathelicidin, LL-37, also synergizes with endogenous inflammatory mediators to enhance the induction of specific inflammatory effects through a complex mechanism involving multiple pathways such as GPCR, EGFR, and TLR [94,95]. As a result, cathelicidin peptides increase cell migration and secretion of cytokines (IL- 6, IL-8, IL-10, IL-18, and IP-10) and chemokines (MCP-1, MIP3α, and RANTES) from activated cells. Also, dermcidin-1L (DCD-1L) stimulates keratinocytes to generate cytokines (TNF- α, IL-8, and IP-10) and chemokines (MIP3α) through both G protein and p38/MAPK pathways [96].
Comment 7:
How does direct and indirect microbial transmission occur among athletes, and what preventive measures can be implemented? (Lines 203–214)
Response 7: Thank you for point this out. We agree with this comment.
There are several mechanisms by which infectious agents may be spread among athletes. Direct contact transmission involves person-to-person contact in which infectious agents are physically transferred from one person to a susceptible host. Indirect-contact transmission occurs when a susceptible host meets contaminated objects or fomites, such as equipment, towels, or clothing. A type of indirect contact is droplet transmission, which occurs when droplets containing infectious agents are generated through coughing, sneezing, or talking and are deposited on the host’s conjunctivae, nasal mucosa, or mouth after being propelled in the air a short distance.
Comment 8:
What impact does exposure to different sports environments (e.g., gym mats, locker rooms) have on the composition and resilience of the skin microbiome? (Lines 208–219)
Response 8: Thank you for point this out. We agree with this comment.
The exposure to different sports environments revealed that skin microbiome composition was enriched with methicillin-susceptible S. aureus (MSSA) and contained methicillin-resistant S. aureus (MRSA) (football locker room and weight room). The fingertip location of S. aureus recovery from football players was significant when compared with both other locations in football players and fingertips in wrestlers and the control group. All S. aureus isolates recovered from athletes in our study were resistant to clindamycin, which is often the drug of choice for soft tissue infections. Trimethoprim/sulfamethoxazole is another drug of choice for soft tissue infections, but half of the isolates we recovered were resistant to it. If the bacteria are resistant to the administered antibiotic, the result may be septicemia; therefore, the clinician should be careful to select an antibiotic to which the infection is susceptible [49-52].
Comment 9:
What are the benefits and limitations of using probiotic and prebiotic interventions to restore and maintain a healthy skin microbiome? (Lines 189–198)
Response 9: Thank you for point this out. We agree with this comment.
The applied microbiota can be (1) alive (probiotics): a probiotic is a living microorganism that, when added in sufficient amounts, exerts a beneficial effect on the host [112]. (2) Tyndallized or thermos-killed bacteria (postbiotics): bacterial cell structures, enzymes and excreted bacterial factors are added, but the bacteria do not replicate anymore. (3) Cell lysates or physically killed bacteria (postbiotics): the bacteria are destroyed, and the cell contents and cell walls are in solution. The bacteria do not replicate anymore, but the enzymes can still be active. (4) Purified enzymes: single or groups of bacterial enzymes are purified and added. (5) Fermentation products or supernatants: the bacteria are not added, but the supernatants containing their antioxidants, amino acids, lipids, and/or vitamins are added. Methods 1–5 have multiple advantages over a skin microbiome transplant with the main advantage being that the process is easily scalable and thus industrial applicable. For method 1 (application of live probiotics), highly concentrated bacteria can be applied; thus, a higher efficacy can be obtained compared to a complete skin microbiome transplant. Pro- or postbiotics can be applied in a skin emollient, creme or suitable medium for skin. There are also a series of drawbacks associated with the use of pro- and postbiotics. Bacteria are cultured in sugar-rich media; it can therefore be more difficult for the bacteria to adjust to a sebum-rich environment. Skin engraftment is not easy; the applied bacteria compete with the skin resident microbiome of the deeper skin layers. The application of high amounts of bacteria could lead to a skin immune reaction with irritation and side effects as a result. A third method of changing the skin microbiome is through prebiotic stimulation. In this process, prebiotics are supplemented to the skin to stimulate the growth of specific health-promoting microbes. A prebiotic is an ingredient with a bioselective activity that exerts a beneficial effect on the host and attempts to improve the host’s health [113]. There are several advantages to this method. There is no need to work with living bacteria; thus, there is a reduced chance of a skin immune reaction. The method has an indirect mechanism of action. Prebiotics are typically well-defined compounds for which side effects are well-studied. The INCI name and safety sheets are normally available. There are also disadvantages. The indirect method has less direct results. Prebiotics could also stimulate non-targeted low-abundance bacteria. The effect of prebiotics can be unpredictable given the variability in the skin microbiome, physiology and immune response in different individuals. All methods have their advantages and disadvantages. Scientific research is currently being conducted using several of these methods to treat common skin disorders.
Comment 10
How does physical exercise modulate the immune response through changes in the skin microbiota, and what implications does this have for infection susceptibility? (Lines 253–263)
Response 10: Thank you for point this out. We agree with this comment.
Physical exercise has been demonstrated to modulate immune function, thereby strengthening the body's defenses against infections interacting with skin microbiota. By modulating the immune system and decreasing levels of pro-inflammatory cytokines, exercise has been observed to alleviate symptoms such as skin thickness and redness, which are common in inflammatory skin diseases. While physical activity confers numerous health benefits, it also presents certain skin, particularly in terms of increased susceptibility to infections [58]. The molecular mechanisms by which physical activity influences skin microbiota are various including improved blood flow to the skin, promotion of skin hydration, positive effects on the hormonal system, increased collagen production in the dermal layer, and positive influences the mechanical properties of the skin [59]. Physical activity stimulates the release of interleukin-15, which prevents mitochondrial dysfunction and promotes mitochondrial biosynthesis. This, in turn, may correlate with better skin hydration in physically active individuals [60-64]. Proper skin hydration improves its natural barrier function, facilitating protection against internal and external irritants, thereby preventing the development of common skin conditions such as atopic dermatitis, contact dermatitis, psoriasis, and acne [60,65,66].
Reviewer 2 Report
Comments and Suggestions for Authors
This manuscript required a revision for improvements
Comments
The manuscript does not explicitly define a clear research gap or hypothesis. It presents an overview but fails to identify critical unresolved questions in the field. Suggested to revise it carefully.
The abstract mentions the effects of probiotics on physical performance, but this aspect is not well-integrated into the rest of the manuscript, making it seem disjointed.
The manuscript does not adequately explore the molecular mechanisms by which physical activity influences skin microbiota. It broadly states correlations but lacks deeper mechanistic explanations.
The manuscript does not maintain a logical flow. It jumps between topics, such as discussing antimicrobial peptides before fully establishing the role of the microbiota. Suggested restructuring the subheadings
The conclusion summarizes the manuscript without providing strong, actionable insights or directions for future research. It should emphasize key findings, knowledge gaps, and recommendations for future studies.
The manuscript contains several grammatical issues that affect clarity and readability.
Example: "How Physical Activity Influence Bacteria" → Incorrect verb conjugation.
After addressing all the comments, this manuscript can be acceptable for further progress
Author Response
Thank you very much for taking the time to review this manuscript. Please find the detailed responses below and the corresponding revisions/corrections highlighted/in track changes in the re-submitted files
Comment 1
The abstract mentions the effects of probiotics on physical performance, but this aspect is not well-integrated into the rest of the manuscript, making it seem disjointed.
Response 1: Thank you for point this out. We agree with this comment.
We have now added more details on the effects of probiotics on physical performance integrating the rest of the manuscript
Comment 2
The manuscript does not adequately explore the molecular mechanisms by which physical activity influences skin microbiota. It broadly states correlations but lacks deeper mechanistic explanations.
Response 2: Thank you for point this out. We agree with this comment.
The molecular mechanisms by which physical activity influences skin microbiota are various including improved blood flow to the skin, promotion of skin hydration, positive effects on the hormonal system, increased collagen production in the dermal layer, and positive influences the mechanical properties of the skin [59]. Physical activity stimulates the release of interleukin-15, which prevents mitochondrial dysfunction and promotes mitochondrial biosynthesis. This, in turn, may correlate with better skin hydration in physically active individuals [60-64]. Proper skin hydration improves its natural barrier function, facilitating protection against internal and external irritants, thereby preventing the development of common skin conditions such as atopic dermatitis, contact dermatitis, psoriasis, and acne [60,65,66].
Comment 3:
The manuscript does not maintain a logical flow. It jumps between topics, such as discussing antimicrobial peptides before fully establishing the role of the microbiota. Suggested restructuring the subheadings
Response 3: Thank you for point this out. We agree with this comment.
We have now restructured the subheadings.
Comment 4:
The conclusion summarizes the manuscript without providing strong, actionable insights or directions for future research. It should emphasize key findings, knowledge gaps, and recommendations for future studies.
Response 4: Thank you for point this out. We agree with this comment.
We have now improved the conclusion.
Comment 5
The manuscript contains several grammatical issues that affect clarity and readability.
Example: "How Physical Activity Influence Bacteria" → Incorrect verb conjugation.
After addressing all the comments, this manuscript can be acceptable for further progress
Response 5: Thank you for point this out. We agree with this comment.
We have now correct verb conjugation.
Reviewer 3 Report
Comments and Suggestions for Authors
Comments to the Authors: Your efforts are appreciated.
Reference number: microorganisms-3536638
Title: Skin Microbiome Overview: How Physical Activity Influence Bacteria.
The authors intended to provide an overview about the skin microbiota and its regulation, focusing our attention on the interplay with physical activity and the impact of probiotics supplementation on physical performance.
Overall comments
* The manuscript is not complete, and not well written. The research should be improved and focus the presented data to highlight how can it add to the subject area.
*Overall, the manuscript included three main sections: Introduction, regulators of skin microbiota, and interactions between physical exercise and skin microbiota. The presented data should be refined. Consequently, the manuscript needs improvement.
*The English editing of the study: Needs improvement.
*Many paragraphs are written without references.
*The name of the organism (such as Staphylococcus aureus, S. epidermidis, R. Mucosa, herpes simplex): should be in italic and should start with capital letters throughout the manuscript.
*The abbreviations (such as C. acnes, S. epidermidis, MRSA, AMPs, herpes simplex, and all the names within table 2) should be used throughout the paper following their initial non-abbreviated appearance.
*The similarity index of the manuscript text is acceptable (25%).
Specific comments
Title: Clear.
Keywords:
- keywords are representative of the research.
- They should be arranged alphabetically.
- I suggest adding (probiotics)
Abstract: Should be improved to reflect the content of the manuscript.
Introduction:
- Not provides sufficient background.
- Not well written.
- Should be improved
The aim of the study: Clearly described in the abstract. The aim of the study should be clearly described in the manuscript itself.
Conclusions and recommendations:
- Should be consistent with the evidence presented.
Tables:
- Table 1: Not clear. Not adequately described. The authors mentioned (data pooled from Grice et al.), but Grice et al. is not included in the references’ list.
- Table 2: Clear. Adequately described.
Figure:
- Clear.
- Adequately described.
References:
- Please, revise the sequence of references within the text. You started by reference number [19]. Many references are missed.
- Many paragraphs are written without references.
- In the references’ list, many references are written without being in the text pf manuscript.
- Lines 139, and 142: [Scudiero et Al.] is not included in the references’ list.
- The available references: Relevant, and most are recent.
- The references should be in the same style and match the journal reference style.
Thanks
Comments on the Quality of English Language*The English editing of the study: Needs improvement.
*Many paragraphs are written without references.
*The name of the organism (such as Staphylococcus aureus, S. epidermidis, R. Mucosa, herpes simplex): should be in italic and should start with capital letters throughout the manuscript.
*The abbreviations (such as C. acnes, S. epidermidis, MRSA, AMPs, herpes simplex, and all the names within table 2) should be used throughout the paper following their initial non-abbreviated appearance.
Author Response
Thank you very much for taking the time to review this manuscript. Please find the detailed responses below and the corresponding revisions/corrections highlighted/in track changes in the re-submitted files
Comment 1:
* The manuscript is not complete, and not well written. The research should be improved and focus the presented data to highlight how can it add to the subject area.
Response 1: Thank you for point this out. We agree with this comment.
The manuscript is now complete, and well written
Comment 2:
*Overall, the manuscript included three main sections: Introduction, regulators of skin microbiota, and interactions between physical exercise and skin microbiota. The presented data should be refined. Consequently, the manuscript needs improvement.
Response 2: Thank you for point this out. We agree with this comment.
The presented data are now refined and the manuscript is improved.
Comment 3: The English editing of the study: Needs improvement.
Response 3: Thank you for point this out. We agree with this comment.
The English editing of the study is now improved.
Comment 4: Many paragraphs are written without references.
Response 4: Thank you for point this out. We agree with this comment.
We have now added references to all paragraphs.
Comment 5: *The name of the organism (such as Staphylococcus aureus, S. epidermidis, R. Mucosa, herpes simplex): should be in italic and should start with capital letters throughout the manuscript.
Response 5: Thank you for point this out. We agree with this comment.
We have now written the name of the organism in italic and start with capital letters throughout the manuscript.
Comment 6: *The abbreviations (such as C. acnes, S. epidermidis, MRSA, AMPs, herpes simplex, and all the names within table 2) should be used throughout the paper following their initial non-abbreviated appearance.
Response 6 : Thank you for point this out. We agree with this comment. Now the abbreviations are used throughout the paper following their initial non-abbreviated appearance.
Comment 7:*The similarity index of the manuscript text is acceptable (25%).
Response 7: Thank you for point this out. Thanks for the comment.
Specific comments
Title: Clear.
Keywords:
- keywords are representative of the research.
- They should be arranged alphabetically.
- I suggest adding (probiotics)
Thank you for point this out. We agree with this comment. We have now arranged keywords alphabetically and have added probiotics.
Abstract: Should be improved to reflect the content of the manuscript.
Thank you for point this out. We agree with this comment. We have improved the abstract to reflect the content of the manuscript
Introduction:
- Not provides sufficient background.
- Not well written.
- Should be improved
Thank you for point this out. We agree with this comment. We have now improved introduction
The aim of the study: Clearly described in the abstract. The aim of the study should be clearly described in the manuscript itself.
Thank you for point this out. We agree with this comment. We have now improved the aim of the study
Conclusions and recommendations:
- Should be consistent with the evidence presented.
Thank you for point this out. We agree with this comment. We have now improved the conclusions.
Tables:
- Table 1: Not clear. Not adequately described. The authors mentioned (data pooled from Grice et al.), but Grice et al. is not included in the references’ list.
- Table 2: Clear. Adequately described.
Thank you for point this out. We agree with this comment. We have now included Grice et al. is not included in the references’ list
Figure:
- Clear.
- Adequately described.
References:
- Please, revise the sequence of references within the text. You started by reference number [19]. Many references are missed.
- Many paragraphs are written without references.
- In the references’ list, many references are written without being in the text pf manuscript.
- Lines 139, and 142: [Scudiero et Al.] is not included in the references’ list.
- The available references: Relevant, and most are recent.
- The references should be in the same style and match the journal reference style.
Thank you for point this out. We agree with this comment. We have now revised the sequence of references within the text.
Round 2
Reviewer 2 Report
Comments and Suggestions for Authors
Accept
Author Response
Thank you very much for taking the time to review this manuscript.
Thanks a lot for the acceptance of the review
Reviewer 3 Report
Comments and Suggestions for Authors
Please, see the attached file.

Author Response
Thank you very much for taking the time to review this manuscript. Please find the detailed responses below and the corresponding revisions/corrections highlighted/in track changes in the re-submitted files
